# The relationship between adherence to continuous positive airway pressure and nasal resistance measured by rhinomanometry in patients with obstructive sleep apnea syndrome

Nobuhiro Fujito, Yasuyoshi Ohshima *, Satoshi Hokari, Atsunori Takahashi, Asuka Nagai, Ryoko Suzuki, Nobumasa Aoki, Satoshi Watanabe, Toshiyuki Koya, Toshiaki Kikuchi

Department of Respiratory Medicine and Infectious Diseases, Niigata University Graduate School of Medical and Dental Sciences, Niigata-shi, Niigata, Japan

* ohshima@med.niigata-u.ac.jp

**Data Availability Statement:** All relevant data are within the paper and its Supporting Information files.

## Abstract

Nasal breathing disorders are associated with obstructive sleep apnea (OSA) syndrome and influence the availability of continuous positive airway pressure (CPAP) therapy. However, information is scarce about the impact of nasal resistance assessed by rhinomanometry on CPAP therapy. This study aimed to examine the relationship between CPAP adherence and nasal resistance evaluated by rhinomanometry, and to identify clinical findings that can affect adherence to CPAP therapy for patients with OSA. This study included 260 patients (199 men, 61 women; age 58 [interquartile ranges (IQR) 50–66] years) with a new diagnosis of OSA who underwent rhinomanometry (before, and 1 and 3 months after CPAP introduction) between January 2011 and December 2018. CPAP use was recorded, and the good and poor CPAP adherence groups at the time of patient registration were compared. Furthermore, those with improved and unimproved pre-CPAP high rhinomanometry values were also compared. Their apnea-hypopnea index (AHI) by polysomnography at diagnosis was 45.6 (IQR 33.7–61.6)/hour, but the residual respiratory event (estimated AHI) at enrollment was 2.5 (IQR 1.4–3.9)/hour and the usage time was 318 (IQR 226–397) minutes, indicating that CPAP was effective and adherence was good. CPAP adherence was negatively correlated with nasal resistance (r = -0.188, p = 0.002). The participants were divided into good (n = 153) and poor (n = 107) CPAP adherence groups. In the poor adherence group, rhinomanometry values before CPAP introduction were worse (inspiration, p = 0.003; expiration, p = 0.006). There was no significant difference in patient background when comparing those with improved (n = 16) and unimproved (n = 12) pre-CPAP high rhinomanometry values. However, CPAP usage time was significantly longer in the improved group 1 month (p = 0.002) and 3 months (p = 0.026) after CPAP introduction. The results suggest that nasal resistance evaluated by rhinomanometry is a useful predictor of CPAP adherence, and that improved rhinomanometry values may contribute to extending the duration of CPAP use.

**Funding:** This work was supported by KAKENHI Grant-in-Aid for Young Scientists (grant no. 17K15824) from the Japan Society for the Promotion of Science (JSPS). Y.O. received the grant. The funders had no role in study design, data collection and analysis, decision to publish, or preparation of the manuscript.

**Competing interests:** The authors have declared that no competing interests exist.

## 1. Introduction

Obstructive sleep apnea (OSA) syndrome is a disease in which the upper airway collapses during sleep, causing frequent and repetitive apnea and hypopnea. Through frequent and repetitive high negative intrathoracic pressure, hypoxia/hypercapnia, and mid-arousal, OSA induces increased sympathetic nerve activity, oxidative stress, inflammation, and vascular endothelial dysfunction. Furthermore, OSA is directly associated with hypertension, cardiovascular disease, and cerebrovascular disease [1, 2], and increases mortality [3–5]. Continuous positive airway pressure (CPAP) is the first-choice treatment for OSA and is recommended for patients with severe forms of the disease [6, 7]. Although CPAP improves cardiovascular parameters of OSA [8], its effectiveness in improving prognosis cannot be noted unless CPAP is used as prescribed [9, 10]. Indeed, adherence to CPAP influences the therapeutic effect in patients with OSA, and CPAP of 4 hours or more per night is ideal to improve subjective daytime sleepiness and reduce the frequency of cardiovascular events [6, 11, 12]. However, it has been reported that 29–83% of patients have poor CPAP adherence [13–15], and improving this factor remains a major issue.

The initial period of CPAP introduction is central for establishing CPAP adherence [16, 17], and adherence during the first month is a predictor of future adherence levels [18]. Furthermore, factors affecting CPAP adherence include patient characteristics (age, nasal cavity volume and nasal resistance, level of understanding of CPAP, self-efficacy, and presence of a bed partner) and medical factors (operability of CPAP equipment, adverse events due to use of CPAP mask interface systems and high pressure during CPAP use, disease requiring long-term treatment [associated with non-radical therapy], CPAP setting and effectiveness, treatment-related factors such as cost, communication between patients and medical staff, patient education, cognitive behavioral therapy, and remote monitoring and/or telephone intervention) [7, 14, 15, 19]. Among these factors, the improvement of nasal resistance by topical treatment contributes to an increase in life quality, but does not ameliorate the apnea-hypopnea index (AHI) [20, 21] or affect the severity of OSA. Furthermore, high nasal resistance complicates CPAP continuation [22, 23]. In particular, nasal resistance of $\geq 0.35$ Pa/cm$^3$/s has been shown to be an independent predictor of a need for nasal surgery [24]. However, it has been reported that nasal resistance is not related to CPAP adherence after 6 months [25], and also that nasal resistance affects CPAP adherence after 1 year [24]. Therefore, there is insufficient information on the relationship between CPAP adherence and nasal resistance in OSA. This retrospective study aims to evaluate nasal resistance, and to identify clinical findings that can affect adherence to CPAP therapy in patients with OSA.

## 2. Methods

We conducted a single-center, retrospective cohort study of patients undergoing CPAP treatment at Niigata University Medical and Dental Hospital between April 2011 and December 2018. Inclusion criteria consisted of adult (age $\geq 20$ years old) patients with a new OSA diagnosis and AHI of $\geq 20$. In Japan, patients with AHI $\geq 20$ are covered by medical insurance for CPAP therapy. Exclusion criteria were absence of nasal resistance by rhinomanometry, poor measurement results with rhinomanometry, discontinuation of CPAP within 3 months, and central sleep apnea (CSA).

Regarding the participants, 397 consecutive patients were registered in 2019. Of these, we excluded 59 who did not undergo nasal ventilation test, 53 with poor measurement results, 24 that discontinued CPAP within 3 months, and one with $\geq 50\%$ CSA. Finally, 260 patients were analyzed.

Polysomnography (PSG) was conducted as an inpatient examination using Somnostar (Sensor Medics, Yorba Linda, CA, USA) or PSG-1100 (Nihon Kohden corp., Tokyo, Japan),

and it comprised electroencephalography, electrooculography, mentalis muscle electromyography, lower extremity electromyography, electrocardiography, body position, arterial blood oxygen saturation ($SpO_2$), oral/nasal airflow, and thoracoabdominal movement. Analysis rules were based on the criteria [26, 27] proposed by the American Academy of Sleep Medicine. Absence of thoracoabdominal movement during apnea was defined as CSA, and CSA <50% among all respiratory events together with AHI ≥ 5 was diagnosed as OSA.

Bilateral nasal resistance (100 Pa) was measured by employing the active anterior method using a nasal aerometer (HI-801; Chest Corp., Tokyo, Japan) during respiration, in the same recumbent position as when the CPAP device was attached. Nasal symptoms were evaluated using the Nasal Symptom Level Classification [28] as a total nasal symptom score, which is the sum of four items: sneezing, nasal discharge, nasal congestion, and disturbance in daily life.

The participants' age, body mass index (BMI), Japanese version of the Epworth Sleepiness Scale (JESS), AHI measured through PSG, arousal index (ArI), and percentage of sleep time spent at $SpO_2$ <90% (CT90%) were evaluated once during diagnosis. Furthermore, the residual respiratory events (estimated AHI; eAHI) detected at CPAP attachment were measured three times: before, and 1 and 3 months after the introduction of CPAP based on the total nasal symptom score and rhinomanometry. Maximum blood pressure, and CPAP use rate and duration estimated by the number of days used out of total days were evaluated before CPAP evaluation, 1 month after CPAP introduction, and upon confirming that the case was registered in 2019.

CPAP adherence was considered to be good [29] when it was used for an average of ≥4 hours and the daily use rate was ≥70%. Based on the CPAP data at the time of case enrollment, the participants were divided into groups corresponding to good and poor CPAP adherence (primary analysis). Furthermore, nasal resistance was considered high when it was ≥0.35 Pa/$cm^3$. Twenty-nine patients whose high pre-CPAP rhinomanometry values (exhalation) raised to >0.35 Pa/$cm^3$/s 3 months after CPAP introduction were considered as the group with improved rhinomanometry values (improvement group). However, those whose rhinomanometry values remained at ≤0.35 Pa/$cm^3$/s 3 months after CPAP introduction were considered as the no-improvement group. Both groups were examined again (secondary analysis).

Statistical values are expressed as medians with IQR, and the Mann-Whitney U test was used to compare the two groups. Spearman's correlation for non-parametric variables was used to identify associations between nasal resistance and CPAP adherence. SPSS version 22 (IBM, Armonk, NY) was used for statistical analyses, with a significance level of 5%.

This study was approved by the Niigata University Ethics Review Board (approval number 2018–0050). A document that specifies the research participants, study period, study purpose, methods used, management of information, contact information, etc., is available on the website of the Niigata University School of Medicine (https://www.med.niigata-u.ac.jp/contents/activity/clinical_research/pdf/2018-0050.pdf). Written informed consent was obtained in the form of opt-out on the website.

## 3. Results

The participants were 199 men and 61 women who were aged 58 (IQR 50–66) years at the time of diagnostic PSG, had a BMI of 26.6 (IQR 24.1–30.7) kg/$m^2$, an AHI of 45.6 (IQR 33.7–61.6)/hour, and were diagnosed with OSA. The rhinomanometry values before the introduction of CPAP were 0.19 (IQR 0.15–0.25) Pa/$cm^3$/s for inhalation, and 0.21 (IQR 0.16–0.27) Pa/$cm^3$/s for exhalation. The rhinomanometry values were high in 21 patients (8.1%) for inhalation, and in 28 patients (10.8%) for exhalation. The total nasal symptom score was 2 (IQR 1–4) points, and 24 patients (9.2%) underwent otolaryngological treatment with oral medications

**Table 1. Patient background by CPAP adherence at enrollment.**

| | ALL | Good adherence group | Poor adherence group | P-value |
|---|---|---|---|---|
| | n = 260 | n = 153 | n = 107 | |
| Sex (Male/Female) | 199/61 | 117/36 | 82/25 | 0.837 |
| Age (years) | 58 (50–66) | 59 (50–66) | 57 (47–65) | 0.071 |
| BMI (kg/m$^2$) | 26.6 (24.1–30.7) | 25.7 (23.8–29.4) | 28.3 (25.0–31.6) | 0.004** |
| JESS (points) | 9 (6–13) | 10 (7–14) | 9 (6–12) | 0.121 |
| Diagnostic PSG test | | | | |
| AHI (/hour) | 45.6 (33.7–61.6) | 42.8 (33.3–66.1) | 46.4 (31.9–58.3) | 0.329 |
| ArI (/hour) | 45.1 (31.5–64.0) | 45.4 (31.2–64.0) | 44.4 (32.8–58.6) | 0.775 |
| CT90% (%) | 9.7 (2.3–27.5) | 8.0 (2.2–27.5) | 13.0 (2.6–26.1) | 0.233 |
| Rhinomanometry (Pa/cm$^3$/s) | | | | |
| Inhalation before CPAP introduction | 0.19 (0.15–0.25) | 0.19 (0.14–0.24) | 0.21 (0.17–0.26) | 0.003** |
| Inhalation 1 month after CPAP introduction | 0.18 (0.15–0.25) | 0.19 (0.14–0.24) | 0.19 (0.15–0.26) | 0.447 |
| Inhalation 3 months after CPAP introduction | 0.18 (0.14–0.23) | 0.18 (0.14–0.23) | 0.20 (0.15–0.25) | 0.048* |
| Exhalation before CPAP introduction | 0.21 (0.16–0.27) | 0.20 (0.15–0.26) | 0.23 (0.17–0.28) | 0.006** |
| Exhalation 1 month after CPAP introduction | 0.20 (0.16–0.27) | 0.20 (0.16–0.27) | 0.21 (0.17–0.28) | 0.405 |
| Exhalation 3 months after CPAP introduction | 0.20 (0.16–0.26) | 0.19 (0.15–0.25) | 0.21 (0.16–0.27) | 0.083 |
| Total nasal symptom score (points) | | | | |
| Before CPAP introduction | 2 (1–4) | 2 (1–3) | 2 (1–4) | 0.417 |
| 1 month after CPAP introduction | 3 (1–5) | 3 (1–5) | 3 (2–5) | 0.420 |
| 3 months after CPAP introduction | 3 (1–5) | 3 (1–4) | 3 (1–5) | 0.473 |
| Otolaryngological treatment (person) | | | | |
| Before CPAP introduction | 24 | 13 | 11 | 0.499 |
| 3 months after CPAP introduction | 35 | 20 | 15 | 0.865 |

There were no significant differences in sex, age, diagnostic PSG results, total nasal symptom score, or otolaryngological treatment. Significant differences were observed in BMI and rhinomanometry values before CPAP introduction.

(**: p < 0.01

*: p < 0.05, Mann-Whitney U test)

BMI, body mass index; JESS, Japanese version of the Epworth Sleepiness Scale; PSG, polysomnography; AHI, apnea-hypopnea index; ArI, Arousal index; CT90%, percentage of sleep time spent at SpO$_2$<90%; CPAP, continuous positive airway pressure

or nasal drops (Table 1). At the time of CPAP introduction, the fixed pressure setting was used in one patient, and the auto setting was used in 259 patients. CPAP treatment was started with a lower pressure limit of 4.0 (IQR 4.0–4.0) cmH$_2$O and an upper pressure limit of 16.0 (IQR 12.0–20.0) cmH$_2$O. CPAP adherence was good in 153 patients (58.8%) who had used CPAP for 817 (IQR 342–1569) days at the time of enrollment, with a use rate of 90.4% (IQR 64.0–98.2) and duration of use of 318 (IQR 226–397) minutes (Table 2). Fifteen patients changed CPAP treatment (11 with poor CPAP adherence), 24 patients discontinued treatment (17 with poor CPAP adherence), and 6 patients died (5 with poor CPAP adherence). CPAP adherence was negatively correlated with nasal resistance measured by rhinomanometry (r = -0.188, p = 0.002). Increased nasal resistance was weakly correlated with lower CPAP use rate, shorter duration of CPAP use, and poor CPAP adherence (Table 3, and Figs 1 and 2).

Patient background characteristics and diagnostic PSG findings were compared between the good (n = 153) and bad (n = 107) CPAP adherence groups at the time of enrollment (Table 1). Obesity was significantly higher (p = 0.004) and rhinomanometry values before CPAP introduction were worse (inhalation, p = 0.003; exhalation, p = 0.006) in the poor adherence group. However, there were no significant differences in sex, age, diagnostic PSG

**Table 2. Changes in CPAP data at enrollment.**

| | ALL | Good adherence group | Poor adherence group | P-value |
|---|---|---|---|---|
| | n = 260 | n = 153 | n = 107 | |
| 1 month after CPAP introduction | | | | |
| Good CPAP adherence | 170 (65.4%) | 129 (84.3%) | 41 (38.3%) | <0.001** |
| CPAP use rate (%) | 95.2 (78.6–100) | 100 (92.0–100) | 78.6 (53.0–96.2) | <0.001** |
| CPAP duration of use (min) | 316 (223–385) | 347 (289–408) | 247 (174–323) | <0.001** |
| eAHI (/hour) | 2.9 (1.5–5.3) | 2.8 (1.6–4.2) | 3.0 (1.5–6.8) | 0.116 |
| Maximum pressure (cmH$_2$O) | 9.8 (7.9–12.0) | 10.0 (8.4–12.2) | 8.9 (7.6–11.9) | 0.046* |
| 3 months after CPAP introduction | | | | |
| Good CPAP adherence | 160 (61.5%) | 129 (84.3%) | 31 (29.0%) | <0.001** |
| CPAP use rate (%) | 93.7 (73.3–100) | 97.4 (89.3–100) | 74.1 (35.0–94.0) | <0.001** |
| CPAP duration of use (min) | 316 (228–391) | 359 (310–409) | 232 (161–311) | <0.001** |
| eAHI (/hour) | 2.4 (1.4–4.2) | 2.2 (1.5–3.7) | 2.8 (1.4–6.2) | 0.069 |
| Maximum pressure (cmH$_2$O) | 9.0 (7.0–11.8) | 9.9 (7.9–11.9) | 8.5 (6.5–11.0) | 0.014* |
| At enrollment | | | | |
| CPAP use rate (%) | 90.4 (64.0–98.2) | 96.6 (91.0–99.9) | 54.9 (27.1–74.7) | <0.001** |
| CPAP duration of use (min) | 318 (226–397) | 369 (324–418) | 204 (152–250) | <0.001** |
| eAHI (/hour) | 2.5 (1.4–3.9) | 2.2 (1.2–3.4) | 2.7 (1.5–4.6) | 0.039* |
| Maximum pressure (cmH$_2$O) | 9.0 (7.5–11.3) | 9.7 (7.9–11.9) | 8.5 (7.0–11.0) | 0.021* |

Significant differences were observed in the rate and duration of use 1 month and 3 months after CPAP introduction.

(**: p < 0.01

*: p < 0.05, Mann-Whitney U test)

CPAP, continuous positive airway pressure; eAHI, estimated apnea-hypopnea index

findings, total nasal symptom score, and presence or absence of otolaryngological treatment. Regarding changes in CPAP adherence, in the poor adherence group, the rate of CPAP use was poor from 1 month after CPAP introduction (p<0.001), the duration of use was short

**Table 3. Correlation between rhinomanometry and CPAP adherence at enrollment.**

| Rhinomanometry | | CPAP adherence | CPAP use rate | CPAP duration of use |
|---|---|---|---|---|
| Inhalation before CPAP introduction | r | -0.188 | -0.159 | -0.176 |
| | p | 0.002** | 0.010** | 0.004** |
| Inhalation 1 month after CPAP introduction | r | -0.047 | -0.032 | -0.057 |
| | p | 0.448 | 0.610 | 0.362 |
| Inhalation 3 months after CPAP introduction | r | -0.123 | -0.074 | -0.077 |
| | p | 0.048* | 0.238 | 0.217 |
| Exhalation before CPAP introduction | r | -0.170 | -0.126 | -0.169 |
| | p | 0.006** | 0.042* | 0.006** |
| Exhalation 1 month after CPAP introduction | r | -0.052 | -0.012 | -0.059 |
| | p | 0.406 | 0.849 | 0.344 |
| Exhalation 3 months after CPAP introduction | r | -0.108 | -0.044 | -0.051 |
| | p | 0.083 | 0.484 | 0.410 |

CPAP adherence was negatively correlated with nasal resistance based on rhinomanometry.

CPAP, continuous positive airway pressure

(**: p < 0.01

*: p < 0.05, Spearman correlation test)

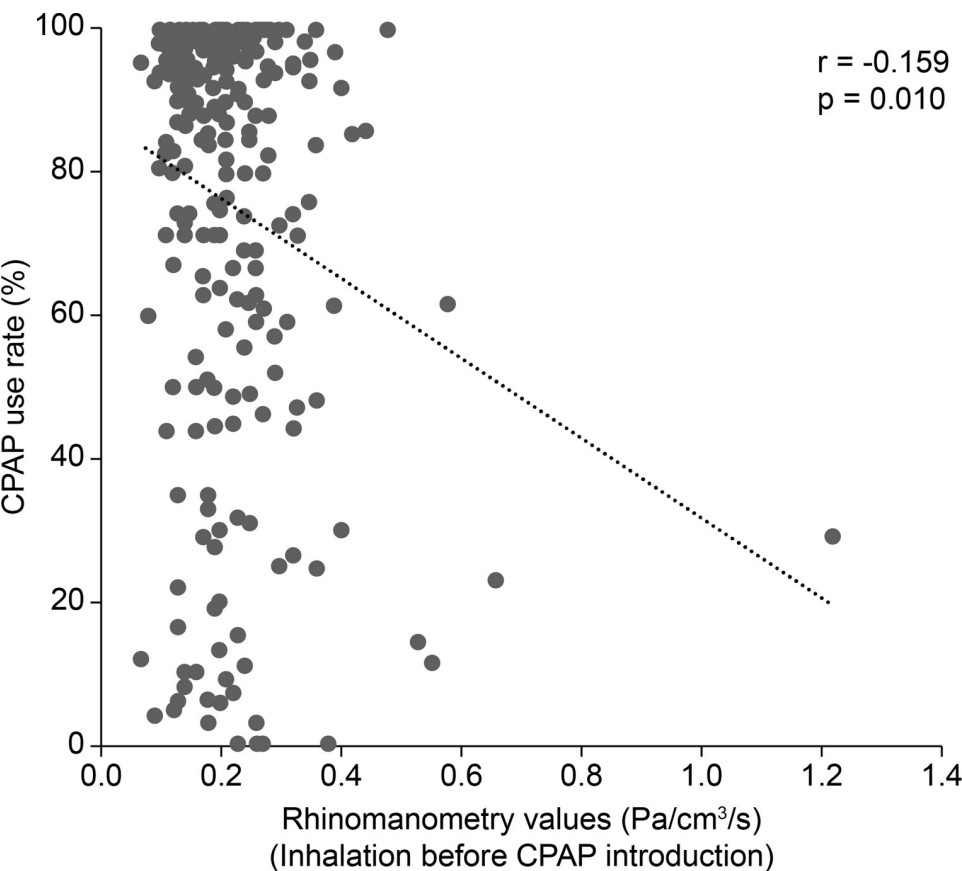

**Fig 1. Relationship between rhinomanometry values before CPAP introduction and CPAP use rate at enrollment.**
Using Spearman's correlation, a weakly correlation was observed between increased nasal resistance and lower CPAP use rate at enrollment. CPAP, continuous positive airway pressure.

(p<0.001), and eAHI was high at enrollment (p = 0.039). In the poor adherence group, the use rate and usage time were lower after 3 months and at enrollment, while in the good adherence group, the use rate and duration of use remained satisfactory (Table 2). When comparing patients with high pre-CPAP introduction rhinomanometry values (exhalation) that improved (n = 16) or did not improve (n = 12), there was no significant difference in their background other than the total nasal symptom score before CPAP introduction (Table 4). However, regarding changes in CPAP data, the CPAP usage time was significantly longer in the group with improved rhinomanometry values at 1 and 3 months after CPAP introduction (p = 0.002 and p = 0.026, respectively) (Table 5).

## 4. Discussion

CPAP treatment for patients with OSA is highly effective whenever adherence is good, and adherence information obtained from CPAP devices is clinically useful. However, information on the relationship between adherence and nasal resistance measured by rhinomanometry is insufficient and needs further investigation. In this study, we longitudinally acquired information on objective CPAP adherence obtained from CPAP devices, and examined its relationship with rhinomanometry values. Satisfactory rhinomanometry values before CPAP introduction were predictors of good CPAP adherence, which was in turn maintained continuously from the time of CPAP introduction. However, there was no significant difference in subjective

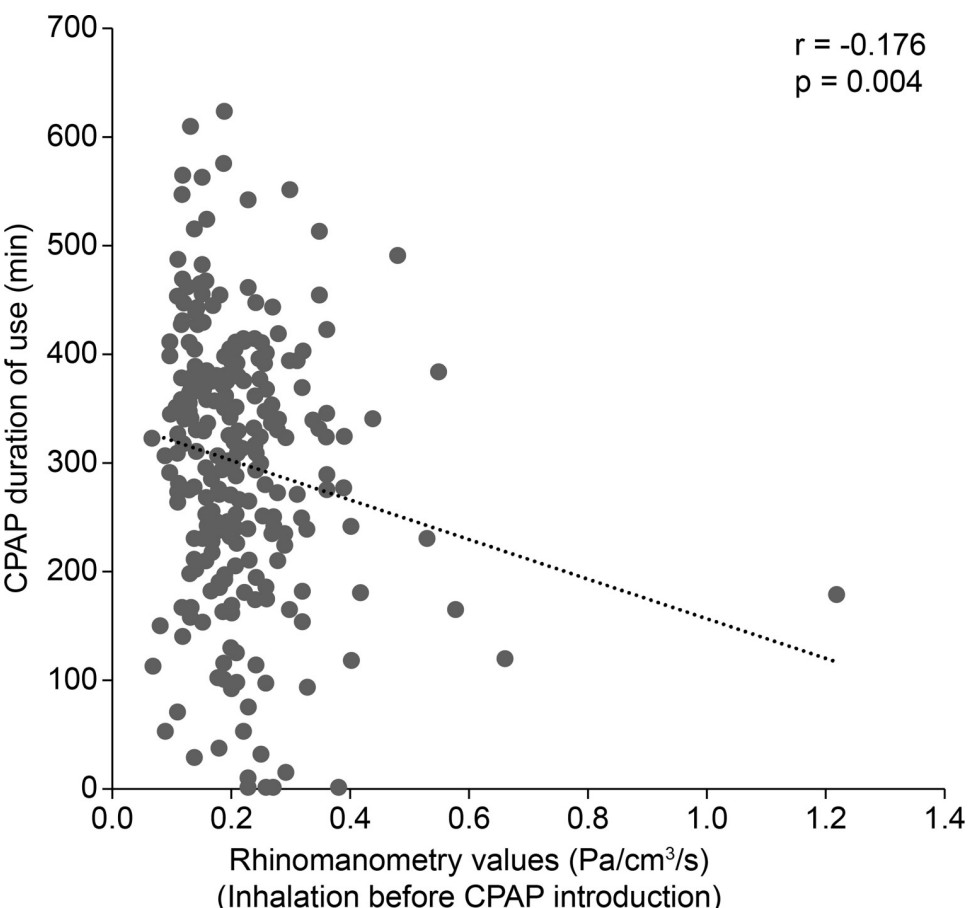

**Fig 2. Relationship between rhinomanometry values before CPAP introduction and CPAP duration of use at enrollment.** Using Spearman's correlation, a weakly correlation was observed between increased nasal resistance and shorter duration of CPAP use at enrollment. CPAP, continuous positive airway pressure.

symptoms such as drowsiness, nasal symptoms, and diagnostic PSG results among the patients.

Rhinomanometry values were good in the group with good CPAP adherence after a median of 2.2 years at the time of enrollment. There are no reports on long-term CPAP adherence and rhinomanometry values, but this finding agrees with a previous report [24] that showed a relationship between CPAP adherence and rhinomanometry values 1 year later. Furthermore, the group with good CPAP adherence at the time of enrollment also showed good CPAP adherence 1 month and 3 months after CPAP introduction. This also agrees with past reports [16–18] stating that adherence in the early stages of CPAP introduction is important for long-term CPAP adherence. Nasal resistance is crucial for good CPAP adherence. However, even if nasal resistance is poor, if the nasal air permeability improves after the introduction of CPAP, it is possible to use CPAP for a significantly longer time in the early stages of CPAP introduction. This suggests that the total nasal symptom score before CPAP introduction may be helpful. The BMI and CT90% tended to be different between the improvement and no improvement groups. A relation between obesity and increased nasal resistance and hypoxia has been suggested [30], and physiological and anatomical factors of the nose may have affected the difference in rhinomanometry results. In this study, otolaryngological intervention was performed in 24 patients (9.2%) before CPAP introduction, and intervention was not performed in 35 patients (13.5%)

**Table 4. Patient background based on the presence or absence of sustained high nasal air permeability (exhalation).**

| | Improvement group | No improvement group | P-value |
|---|---|---|---|
| | n = 16 | n = 12 | |
| Sex (Male/Female) | 14/2 | 11/1 | 0.873 |
| Age (years) | 63.5 (58–67) | 50 (46.5–60.5) | 0.260 |
| BMI (kg/m$^2$) | 24.9 (23.5–30.5) | 29.1 (26.1–31.4) | 0.082 |
| JESS (points) | 11.5 (10–14) | 11 (8.5–17) | 0.867 |
| Diagnostic PSG | | | |
| AHI (/hour) | 47.4 (41.3–78.9) | 45.5 (24.2–73.4) | 0.664 |
| ArI (/hour) | 53.5 (36.8–81.0) | 47.1 (29.1–72.7) | 0.837 |
| CT90% (%) | 4.8 (2.4–22.0) | 29.1 (9.4–45.5) | 0.133 |
| Rhinomanometry (Pa/cm$^3$/s) | | | |
| Before CPAP introduction | 0.41 (0.38–0.44) | 0.44 (0.37–0.62) | 0.189 |
| 1 month after CPAP introduction | 0.29 (0.26–0.40) | 0.41 (0.29–0.57) | 0.189 |
| 3 months after CPAP introduction | 0.22 (0.20–0.27) | 0.44 (0.38–0.51) | <0.001** |
| Nasal symptoms score (points) | | | |
| Before CPAP introduction | 3 (2–3) | 5 (4–6.5) | 0.037* |
| 1 month after CPAP introduction | 2 (1–5) | 5 (3–6) | 0.260 |
| 3 months after CPAP introduction | 2 (1–5) | 5 (4–5) | 0.121 |
| Otolaryngological treatment | | | |
| Before CPAP introduction | 2 | 3 | 0.599 |
| 3 months after CPAP introduction | 2 | 4 | 0.371 |

There were no significant differences in sex, age, BMI, diagnostic PSG results, or otolaryngological treatment. Significant differences were observed in the total nasal symptom score before CPAP introduction.

(**: p < 0.01

*: p < 0.05, Mann-Whitney U test)

BMI, body mass index; JESS, Japanese version of the Epworth Sleepiness Scale; PSG, polysomnography; AHI, apnea-hypopnea index; ArI, Arousal index; CT90%, percentage of sleep time spent at SpO$_2$<90%; CPAP, continuous positive airway pressure

even 3 months after CPAP introduction. In line with this, appropriate intervention for nasal breathing disorders is recommended in the early stages of CPAP introduction [31]. Nevertheless, it has been reported that there is no significant difference in CPAP adherence to placebo or humidifier use, regardless of the presence of nasal symptoms when CPAP is introduced, even after treatment with steroid nasal drops [32, 33]. Thus, it is important to select appropriate cases and to work closely with the otorhinolaryngology department.

Furthermore, we also investigated factors other than rhinomanometry nasal resistance that affect CPAP adherence. There was no difference in patient sleepiness or diagnostic PSG results between the groups, therefore prediction of CPAP adherence before treatment was difficult. Both sleepiness and OSA severity are considered factors that weakly affect CPAP adherence when acting alone [14]. However, given that CPAP adherence of the included patients was related to various additional factors including nasal resistance, it may not be necessary to further consider them. In addition, the total nasal symptom score tended to be higher in the poor CPAP adherence group, although the difference was not significant. It has also been suggested that subjective nasal congestion [34] and discharge [35] affect CPAP adherence. However, nasal symptoms alone may not play a significant role in CPAP adherence, as various factors such as nasal mask skin irritation, dry mouth, air leakage, and mask discomfort might be involved [36]. We believe that nasal symptoms should be considered a weak predictor of CPAP adherence.

**Table 5. Changes in CPAP data between the groups with and without persistent high rhinomanometry values (exhalation).**

| | Improvement group | No improvement group | P-value |
|---|---|---|---|
| | n = 16 | n = 12 | |
| One month after CPAP introduction | | | |
| Good CPAP adherence | 13 (81.3%) | 7 (58.3%) | 0.324 |
| CPAP use rate (%) | 96.6 (87.5–100) | 76.2 (65.3–92.5) | 0.047* |
| CPAP duration of use (min) | 339 (321–393) | 263 (232–295) | 0.002** |
| eAHI (/hour) | 3.0 (1.7–5.4) | 2.6 (2.0–3.7) | 0.909 |
| Maximum pressure (cmH$_2$O) | 10.3 (8.0–11.5) | 9.9 (7.2–13.0) | 0.982 |
| Three months after CPAP introduction | | | |
| Good CPAP adherence | 12 (75.0%) | 5 (41.7%) | 0.146 |
| CPAP use rate (%) | 96.5 (78.6–100) | 73.3 (47.0–93.3) | 0.026* |
| CPAP duration of use (min) | 336 (284–396) | 252 (220–328) | 0.026* |
| eAHI (/hour) | 1.9 (1.4–4.6) | 2.1 (1.4–3.0) | 0.942 |
| Maximum pressure (cmH$_2$O) | 10.1 (6.8–11.5) | 8.5 (7.5–12.0) | 0.716 |
| At enrollment | | | |
| Good CPAP adherence | 9 (56.3%) | 4 (33.3%) | 0.324 |
| CPAP use rate (%) | 88.9 (48.3–98.3) | 61.6 (27.1–84.9) | 0.133 |
| CPAP duration of use (min) | 306 (240–342) | 180 (165–354) | 0.347 |
| eAHI (/hour) | 2.4 (1.7–3.4) | 2.5 (2.0–3.2) | 0.399 |
| Maximum pressure (cmH$_2$O) | 9.7 (7.4–12.0) | 7.2 (7.0–7.9) | 0.152 |

There was a significant difference in CPAP usage time 1 month and 3 months after CPAP introduction.

(**: $p < 0.01$

*: $p < 0.05$, Mann-Whitney U test)

CPAP, continuous positive airway pressure; eAHI, estimated apnea-hypopnea index

The poor CPAP adherence group showed higher BMI and higher eAHI in the baseline data. The poor adherence group might be affected by the inclusion of the phenotype, which was characterized by severe OSA, high BMI (28–38 kg/m$^2$), low hypoxemia, ESS score (9–10), and low CPAP adherence [37]. In addition, if the effect of CPAP is insufficient and the residual AHI is high, the patient cannot experience the therapeutic effect of CPAP, which in turn favors poor CPAP adherence [19, 38]. Therefore, monitoring the therapeutic effect is crucial. In Japan, remote monitoring of CPAP therapy was newly established in the 2018 medical fee revision. CPAP remote monitoring guidance is expected to improve CPAP adherence, reduce the burden on medical staff, and improve convenience for patients [7].

This study had some limitations that should be acknowledged. First, it was a retrospective study; second, it was conducted in a single facility; third, no nasal examination or anatomical findings were obtained, with nasal breathing disorders evaluated only by the nasal airflow test and total nasal symptom score; and fourth, manual titration was not performed and the setting pressure was changed depending on the patient's poor adherence status, which may have affected adherence. Furthermore, we evaluated nasal resistance by rhinomanometry in patients with OSA in the recumbent position, who were awake during the day without CPAP. To our knowledge, no method has been established to measure nasal resistance during sleep or CPAP treatment, and determining a more reliable measurement method is essential. Thus, in conjunction with the department of otorhinolaryngology, we aim to conduct a prospective study at multiple centers, examining the effect of various factors on improving CPAP adherence.

## 5. Conclusion

We investigated the relationship between CPAP adherence and rhinomanometry values in 260 patients with OSA. The rhinomanometry values of the CPAP adherence group were significantly better, suggesting that assessing nasal resistance by rhinomanometry is useful for predicting future CPAP adherence, and that, if nasal air permeability can be improved, may contribute to extending the duration of CPAP use. To improve CPAP adherence we believe that close cooperation with the otolaryngology department is required, in order to monitor both CPAP treatment effects and adherence.

## Supporting information

**S1 Checklist. STROBE statement—checklist of items that should be included in reports of observational studies.**
(DOCX)

**S1 Table. Anonymized study dataset.** The relationship between adherence to continuous positive airway pressure and nasal resistance measured by rhinomanometry in patients with obstructive sleep apnea syndrome.
(XLSX)

## Acknowledgments

The authors would like to thank Tazuko Kikuya (Niigata University) for her assistance in acquiring and collecting data for this study.

We would like to thank Editage (www.editage.com) for English language editing.

## Author Contributions

**Conceptualization:** Yasuyoshi Ohshima.

**Data curation:** Nobuhiro Fujito, Yasuyoshi Ohshima.

**Formal analysis:** Nobuhiro Fujito, Yasuyoshi Ohshima.

**Funding acquisition:** Yasuyoshi Ohshima.

**Investigation:** Nobuhiro Fujito, Yasuyoshi Ohshima, Satoshi Hokari, Atsunori Takahashi, Asuka Nagai, Ryoko Suzuki.

**Methodology:** Nobuhiro Fujito, Yasuyoshi Ohshima.

**Project administration:** Yasuyoshi Ohshima.

**Resources:** Yasuyoshi Ohshima, Satoshi Hokari, Ryoko Suzuki.

**Software:** Yasuyoshi Ohshima.

**Supervision:** Toshiyuki Koya, Toshiaki Kikuchi.

**Validation:** Satoshi Hokari, Nobumasa Aoki, Satoshi Watanabe.

**Visualization:** Nobuhiro Fujito, Yasuyoshi Ohshima.

**Writing – original draft:** Nobuhiro Fujito, Yasuyoshi Ohshima.

**Writing – review & editing:** Satoshi Hokari, Asuka Nagai.

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
