## [Decision Letter · Decision Letter 0]

5 Feb 2023

PONE-D-22-28859The relationship between adherence to continuous positive airway pressure and nasal resistance measured by rhinomanometry in patients with obstructive sleep apnea syndromePLOS ONE

Dear Dr. Ohshima,

Thank you for submitting your manuscript to PLOS ONE. After careful consideration, we feel that it has merit but does not fully meet PLOS ONE’s publication criteria as it currently stands. Therefore, we invite you to submit a revised version of the manuscript that addresses the points raised during the review process.

We look forward to receiving your revised manuscript.

Kind regards,

Hyungchae Yang, M.D. & Ph.D

Academic Editor

PLOS ONE

Journal Requirements:

Reviewers' comments:

Reviewer's Responses to Questions

**Comments to the Author**

1. Is the manuscript technically sound, and do the data support the conclusions?

Reviewer #1: Yes

2. Has the statistical analysis been performed appropriately and rigorously? 

Reviewer #1: I Don't Know

3. Have the authors made all data underlying the findings in their manuscript fully available?

Reviewer #1: Yes

4. Is the manuscript presented in an intelligible fashion and written in standard English?

Reviewer #1: Yes

5. Review Comments to the Author

Reviewer #1: In this study, the authors examined the relationship between nasal resistance measured by rhinomanometry and CPAP adherence. I believe that identifying the key factors related to CPAP adherence is an important task in the treatment of sleep apnea and that nasal resistance is one of the key factors that determines this. This paper also shows that nasal resistance may have an effect on CPAP adherence. However, I have some questions.

1. The participants in the study were divided into groups of good and poor CPAP adherence based on their CPAP usage results. Then, the study analyzed the CPAP results and rhinomanometry results after 1 month, 3 months, and at the time of inclusion in the study. I am curious to know whether all of the participants included in the study are currently using CPAP or if there are also those who have discontinued it. I believe that whether or not patients have discontinued or continued using CPAP, separate from the CPAP parameters, is also an important perspective in evaluating CPAP adherence. And, if possible, it would be good to also show the continuity of CPAP at the time of patient enrollment in the study.

2. I'm curious about the accurate meaning of "213 line's Satisfactory rhinomanometry values." Although the presented table shows that the difference of rhinomanometry values are statistically significant, there doesn't seem to be a big difference in the presented median values. How does the author evaluate rhinomanometry results as satisfactory? What specific value is used as a criterion for predicting CPAP adherence? Additionally, it would be helpful to understand the relationship between rhinomanometry values and CPAP adherence by showing a scatter plot.

3. What is the reason the author thinks that nasal resistance is higher in the Poor adherence group compared to the Good adherence group, but CPAP max pressure is lower in the Poor adherence group?

4. The Poor adherence group shows a higher BMI. How does the author think BMI affects adherence?

5. There is a significant difference in terms of Age, BMI, CT90% between the improvement group and the no improvement group, and Nasal symptom score also consistently high in the no improvement group. Would it be possible that the physiological or anatomical factors of the nose have affected the difference in rhinomanometry results?

6. Can you please provide a more detailed description of what the presented CPAP use rate(%) and CPAP duration of use(min) mean in the paper? Are they measuring adherence by the proportion of usage for at least 4 hours and the number of days used out of total days? Additionally, it would be helpful if additional parameters for evaluating CPAP adherence were also presented.

7. What is the reason for setting the OSA inclusion criteria to AHI 20 or higher?

6. PLOS authors have the option to publish the peer review history of their article (what does this mean?). If published, this will include your full peer review and any attached files.

Reviewer #1: No

---

## [Author Response · Author response to Decision Letter 0]

17 Feb 2023

PONE-D-22-28859

The relationship between adherence to continuous positive airway pressure and nasal resistance measured by rhinomanometry in patients with obstructive sleep apnea syndrome

Dear Dr. Hyungchae Yang,

We are resubmitting the above-cited manuscript, which has been revised in accordance with the Reviewer’s helpful comments. We attempted to address the issues raised by the Reviewer, and we have carefully corrected any grammatical errors. Our point-by-point responses to the Reviewer’s comments are provided below.

Response to Reviewers

Reviewer #1: In this study, the authors examined the relationship between nasal resistance measured by rhinomanometry and CPAP adherence. I believe that identifying the key factors related to CPAP adherence is an important task in the treatment of sleep apnea and that nasal resistance is one of the key factors that determines this. This paper also shows that nasal resistance may have an effect on CPAP adherence. However, I have some questions.

(Reply)

Thank you very much for your comment. We have carefully considered all of your questions, and we provide our responses below. 

1. The participants in the study were divided into groups of good and poor CPAP adherence based on their CPAP usage results. Then, the study analyzed the CPAP results and rhinomanometry results after 1 month, 3 months, and at the time of inclusion in the study. I am curious to know whether all of the participants included in the study are currently using CPAP or if there are also those who have discontinued it. I believe that whether or not patients have discontinued or continued using CPAP, separate from the CPAP parameters, is also an important perspective in evaluating CPAP adherence. And, if possible, it would be good to also show the continuity of CPAP at the time of patient enrollment in the study.

(Reply)

Thank you for bringing this oversight to our attention. We have added the following sentence in the revised manuscript:

(Page 9, Lines 149-151)

“Fifteen patients changed CPAP treatment (11 with poor CPAP adherence), 24 patients discontinued treatment (17 with poor CPAP adherence), and 6 patients died (5 with poor CPAP adherence).”

2. I'm curious about the accurate meaning of "213 line's Satisfactory rhinomanometry values." Although the presented table shows that the difference of rhinomanometry values are statistically significant, there doesn't seem to be a big difference in the presented median values. How does the author evaluate rhinomanometry results as satisfactory? What specific value is used as a criterion for predicting CPAP adherence? Additionally, it would be helpful to understand the relationship between rhinomanometry values and CPAP adherence by showing a scatter plot.

(Reply)

We appreciate your insightful question. The scatter plots are indeed very helpful. We have therefore included them in the revised manuscript as Figures 1 and 2.

As you suggested, the difference in rhinomanometry values is small, and the statistical cutoff value is 0.165; however, we believe that it is clinically informative.

(Figure 1 legend)

“Relationship between rhinomanometry values before CPAP introduction and CPAP use rate at enrollment.”

(Figure 2 legend)

“Relationship between rhinomanometry values before CPAP introduction and CPAP duration of use at enrollment.”

3. What is the reason the author thinks that nasal resistance is higher in the Poor adherence group compared to the Good adherence group, but CPAP max pressure is lower in the Poor adherence group?

(Reply)

Thank you for your question. This relates to the change of settings to improve adherence in cases of poor adherence, by lowering the maximum pressure when CPAP use was difficult due to high pressure. We have added the following sentence in the limitations section of the revised manuscript:

(Page 28, Lines 283-285)

“and fourth, manual titration was not performed and the setting pressure was changed depending on the patient's poor adherence status, which may have affected adherence.”

4. The Poor adherence group shows a higher BMI. How does the author think BMI affects adherence?

(Reply)

Thank you for your question. 

There are various studies on the relationship between CPAP adherence and BMI, including some that find no association [Chest. 2002;121:430-435., Sleep. 2007;30:320-324.], others that report a U-shaped association [Sleep Medicine. 2018;51:85-91., Respiration. 2014;87:121-128.], and others that find that the higher the BMI the more sleepiness the patient and the more likely they were to experience improvement in sleepiness when treated with CPAP, thus improving adherence. In particular, a meta-analysis shows that BMI ≥30 kg/m2 and ESS score ≥11 can be expected to show stable adherence based on sleepiness improvement [Front Neurol. 2022;27;13:911996.].

Based on an average BMI of 26.6 and an average ESS of 9 in this study, it is difficult to consider the association between BMI and adherence, which we speculate is complicated by the existence of various OSA phenotypes [Chest. 2020;157:403-420.]. Among the various phenotypes, subtypes with severe OSA (AHI, 34-68), high BMI (28-38 kg/m2), low hypoxemia for a given AHI (CT90% 0-12%), and ESS score (9-10) are defined. The pharyngeal collapsibility, low arousal threshold, and/or elevated loop gain may contribute to pathogenesis in this subtype, which has the lowest CPAP use rate. We believe that the poor adherence group may be affected by the inclusion of the second most common PSG phenotype, which is characterized by severe OSA, low hypoxemia, and low CPAP adherence.

We have added the following sentence and a relevant reference in the revised manuscript:

(Page 27, Lines 272-274)

“The poor adherence group might be affected by the inclusion of the phenotype, which was characterized by severe OSA, high BMI (28-38 kg/m2), low hypoxemia, ESS score (9-10), and low CPAP adherence [37]. In addition,”

(References)

“37. Zinchuk A, Yaggi HK. Phenotypic subtypes of OSA: A challenge and opportunity for precision medicine. Chest. 2020;157: 403–420. doi: 10.1016/j.chest.2019.09.002.”

5. There is a significant difference in terms of Age, BMI, CT90% between the improvement group and the no improvement group, and Nasal symptom score also consistently high in the no improvement group. Would it be possible that the physiological or anatomical factors of the nose have affected the difference in rhinomanometry results?

(Reply)

Thank you for your question. We do think that the physiological or anatomical factors of the nose may have affected the difference in rhinomanometry results. We have added the following sentence and a relevant reference in the revised manuscript:

(Pages 25-26, Lines 246-249)

“The BMI and CT90% tended to be different between the improvement and no improvement groups. A relation between obesity and increased nasal resistance and hypoxia has been suggested [30], and physiological and anatomical factors of the nose may have affected the difference in rhinomanometry results.”

(References)

“30. Tagaya M, Nakata S, Yasuma F, Noda A, Morinaga M, Yagi H, et al. Pathogenetic role of increased nasal resistance in obese patients with obstructive sleep apnea syndrome. Am J Rhinol Allergy. 2010;24: 51–54. doi: 10.2500/ajra.2010.24.3382.”

6. Can you please provide a more detailed description of what the presented CPAP use rate(%) and CPAP duration of use(min) mean in the paper? Are they measuring adherence by the proportion of usage for at least 4 hours and the number of days used out of total days? Additionally, it would be helpful if additional parameters for evaluating CPAP adherence were also presented.

(Reply)

Thank you for bringing this oversight to our attention. The relevant sentence has been rewritten as follows:

(Page 7, Line 112)

“estimated by the number of days used out of total days”

7. What is the reason for setting the OSA inclusion criteria to AHI 20 or higher?

(Reply)

Thank you for your question. This is because of Japanese insurance coverage. We have added the following sentence in the revised manuscript:

(Page 6, Lines 85-86)

“In Japan, patients with AHI ≥20 are covered by medical insurance for CPAP therapy.”

---

## [Decision Letter · Decision Letter 1]

2 Mar 2023

The relationship between adherence to continuous positive airway pressure and nasal resistance measured by rhinomanometry in patients with obstructive sleep apnea syndrome

PONE-D-22-28859R1

Dear Dr. Ohshima,

We’re pleased to inform you that your manuscript has been judged scientifically suitable for publication and will be formally accepted for publication once it meets all outstanding technical requirements.

Kind regards,

Hyungchae Yang, M.D. & Ph.D

Academic Editor

PLOS ONE

Reviewers' comments:

Reviewer's Responses to Questions

**Comments to the Author**

1. If the authors have adequately addressed your comments raised in a previous round of review and you feel that this manuscript is now acceptable for publication, you may indicate that here to bypass the “Comments to the Author” section, enter your conflict of interest statement in the “Confidential to Editor” section, and submit your "Accept" recommendation.

Reviewer #1: All comments have been addressed

2. Is the manuscript technically sound, and do the data support the conclusions?

Reviewer #1: Yes

3. Has the statistical analysis been performed appropriately and rigorously? 

Reviewer #1: Yes

4. Have the authors made all data underlying the findings in their manuscript fully available?

Reviewer #1: Yes

5. Is the manuscript presented in an intelligible fashion and written in standard English?

Reviewer #1: Yes

6. Review Comments to the Author

Reviewer #1: (No Response)

7. PLOS authors have the option to publish the peer review history of their article (what does this mean?). If published, this will include your full peer review and any attached files.

Reviewer #1: No

---

## [Editor Report · Acceptance letter]

6 Mar 2023

PONE-D-22-28859R1 

The relationship between adherence to continuous positive airway pressure and nasal resistance measured by rhinomanometry in patients with obstructive sleep apnea syndrome 

Dear Dr. Ohshima:

I'm pleased to inform you that your manuscript has been deemed suitable for publication in PLOS ONE. Congratulations! Your manuscript is now with our production department. 

Kind regards, 

on behalf of

Dr. Hyungchae Yang 

Academic Editor

PLOS ONE